# Multidisciplinary Approach to the Diagnosis of Idiopathic Interstitial Pneumonias: Focus on the Pathologist’s Key Role

**DOI:** 10.3390/ijms25073618

**Published:** 2024-03-23

**Authors:** Stefano Lucà, Francesca Pagliuca, Fabio Perrotta, Andrea Ronchi, Domenica Francesca Mariniello, Giovanni Natale, Andrea Bianco, Alfonso Fiorelli, Marina Accardo, Renato Franco

**Affiliations:** 1Pathology Unit, Department of Mental and Physical Health and Preventive Medicine, Università degli Studi della Campania “Luigi Vanvitelli”, 80138 Naples, Italy; stefano.luca@unicampania.it (S.L.); francesca.pagliuca@unicampania.it (F.P.); andrea.ronchi@unicampania.it (A.R.); marina.accardo@unicampania.it (M.A.); 2Department of Translational Medical Science, Università degli Studi della Campania “Luigi Vanvitelli”, 80131 Naples, Italy; fabio.perrotta@unicampania.it (F.P.); domenicafrancesca.mariniello@unicampania.it (D.F.M.); andrea.bianco@unicampania.it (A.B.); 3Division of Thoracic Surgery, Università degli Studi della Campania “Luigi Vanvitelli”, Piazza Miraglia, 2, 80138 Naples, Italy; giovanni.natale@unicampania.it (G.N.); alfonso.fiorelli@unicampania.it (A.F.)

**Keywords:** interstitial lung disease, idiopathic interstitial pneumonias, idiopathic pulmonary fibrosis, usual interstitial pneumonia, nonspecific interstitial pneumonia, organizing pneumonia, diffuse alveolar damage, desquamative interstitial pneumonia, pleuroparenchymal fibroelastosis, multidisciplinary discussion

## Abstract

Idiopathic Interstitial Pneumonias (IIPs) are a heterogeneous group of the broader category of Interstitial Lung Diseases (ILDs), pathologically characterized by the distortion of lung parenchyma by interstitial inflammation and/or fibrosis. The American Thoracic Society (ATS)/European Respiratory Society (ERS) international multidisciplinary consensus classification of the IIPs was published in 2002 and then updated in 2013, with the authors emphasizing the need for a multidisciplinary approach to the diagnosis of IIPs. The histological evaluation of IIPs is challenging, and different types of IIPs are classically associated with specific histopathological patterns. However, morphological overlaps can be observed, and the same histopathological features can be seen in totally different clinical settings. Therefore, the pathologist’s aim is to recognize the pathologic–morphologic pattern of disease in this clinical setting, and only after multi-disciplinary evaluation, if there is concordance between clinical and radiological findings, a definitive diagnosis of specific IIP can be established, allowing the optimal clinical–therapeutic management of the patient.

## 1. Introduction

Interstitial Lung Disease (ILD) is an umbrella term used for a large and heterogenous group of diseases affecting the lung parenchyma with inflammation and/or fibrosis [1,2]. The pulmonary interstitium is the site where histopathological abnormalities initially develop, thus explaining how the word “interstitial” was chosen. Nevertheless, several types of ILDs later cause significant changes to the alveolar and airway architecture. The clinical consequence is impaired gas exchange, resulting in breathlessness, diminished exercise tolerance, and impairment of lung function and quality of life [3]. Unlike neoplastic lung disease [4,5], inflammatory lung disease is often a source of great diagnostic difficulties for pathologists. Nonetheless, a precise diagnosis is essential because the prognosis and course of treatment differ greatly amongst ILDs. Although spontaneous reversibility or stabilization of the disease could be a possible evolution, progressive and irreversible pulmonary fibrosis leading to respiratory failure and death can occur [6,7,8]. Currently, ILDs are classified into five categories (Figure 1), each including defined clinical entities, characterized by specific clinical–radiological, histopathological and prognostic features, with a different proportion of cases manifesting progressive pulmonary fibrosis (PPF) [9,10]. Idiopathic Interstitial Pneumonias (IIPs) are a heterogeneous subset of the broader category of ILDs, characterized by unknown etiology and a distortion of lung parenchyma by variable combinations of interstitial inflammation and fibrosis [11]. Over the years, the classification of IIPs has been revised [12], but some disarray characterizes it. An essential advance in their classification occurred in 2002 with the publication of the first American Thoracic Society (ATS)/European Respiratory Society (ERS) international multidisciplinary consensus classification of the IIPs [13], in which the notion of clinical–radiological–pathological diagnosis was emphasized. In 2013, the same group updated the classification, with only a few changes [14]. The main entities proposed in 2002 were preserved, and most histologic patterns previously described were still classified as major IIPs. Lymphoid Interstitial Pneumonia (LIP) and a newly recognized Pleuroparenchymal Fibroelastosis (PPFE) were listed as rare IIPs, and two rare histological patterns (Acute Fibrinous and Organizing Pneumonia and Bronchiolocentric Patterns of Interstitial Pneumonia) were defined (Table 1).

## 2. Diagnostic Approach to Idiopathic Interstitial Pneumonias: A Challenge in Pathology Practice

The Secondary Pulmonary Lobule (SPL) is the smallest and most macroscopically identifiable unit of the lung, measuring 1 to 2 cm. The SPL, including several Primary Lobules, is supplied by a bronchiolovascular bundle consisting of a preterminal bronchiole and a branch of the pulmonary artery [15,16,17]. The SPL is surrounded incompletely by fibrous interlobular septa, extending inward the lung from the visceral pleural surface and including pulmonary vein branches and lymphatic vessels. Veins and lymphatic vessels running inside the secondary lobule are usually located between two primary pulmonary lobules, defined as the pulmonary parenchyma supplied by a respiratory bronchiole and consisting of a small unit of alveolar ducts and alveolar sacs [15,16] (Figure 2A). In the evaluation of IPP, some critical histological areas should be considered: (a) the space around the lobular bronchiole and its artery branch, called the centrilobular area; (b) the centriacinar area, the space around the respiratory bronchioles; (c) the interlobular septa—areas of lung parenchyma adjacent to the visceral pleura; (d) the large bronchovascular bundles and pulmonary veins, representing the periphery of the lobules [18]. Finally, the key element in the pathological evaluation of IIPs is the concept of lung interstitium. The bronchovascular/bronchiolovascular bundles, the alveolar duct/sac, the stroma of the interlobular septa and the subpleural stroma are all part of the lung’s interstitium, also known as the interstitial space, a fibrous stroma made up of collagen and elastic fibers [19]. The lung interstitium is divided into three connected compartments: the axial, parenchymal (also known as alveolar), and peripheral (also known as subpleural) spaces. The axial interstitium corresponds to peribronchovascular and peribronchiolovascular stroma, continuous with the interlobular septa; the alveolar interstitium is the stroma located around the alveolar duct/sac; the peripheral or subpleural interstitium is the stroma at the boundary between the lung parenchyma and visceral pleura, offering a supportive framework to the lung [20,21]. The concepts of lung interstitium and secondary pulmonary lobule are essential to the pathological evaluation of IIPs. The major challenge related to IIPs, and ILDs in general, concerns the multidisciplinary approach required for the diagnosis and the morphological, clinical–radiological and physiological overlaps between different entities [18,22]. Furthermore, the identification of IIPs is an exclusion diagnosis that must take into account morphological similarities with non-idiopathic ILDs. Leslie suggests an approach based on the identification of the dominant histological lesion of disease in the specimen [23,24]. Thus, the six dominant histopathological features should be recognized:Acute lung injury;Fibrosis;Cellular infiltrates;Alveolar filling;Nodules—small or large, single or multiple;Minimal changes.

Once the dominant lesions are identified, their exact location and distribution should be noted, at both the macroscopic and the microscopic level. In this area, imaging techniques, mainly high-resolution CT, provide crucial support, answering questions such as: Are the lesions mainly located in the upper part or in the base of the lung? Are the lesions bilateral? Are the lesions diffuse or partial? Do the lesions affect the center or periphery of the lung [25]? On the other hand, a microscopic examination of the lung biopsy assesses the distribution of histopathological lesions within the secondary pulmonary lobule. Pathologists need to determine whether the SPL’s involvement is diffuse or focal/patchy, and if the lesions mostly exhibit a peripheral distribution, such as a major subpleural and/or paraseptal arrangement, or an airway-centered or bronchiolocentric distribution, i.e., primarily located around the respiratory bronchioles. These findings define the pathologic–morphologic patterns of the IIPs, as well as some essential diagnostic histopathological concepts, such as “temporal heterogeneity”, “temporal homogeneity/uniformity”, “spatial heterogeneity” and “spatial homogeneity/uniformity”, which are pathognomonic of some pathologic–morphologic patterns [18,22].

**Figure 2 ijms-25-03618-f002:**
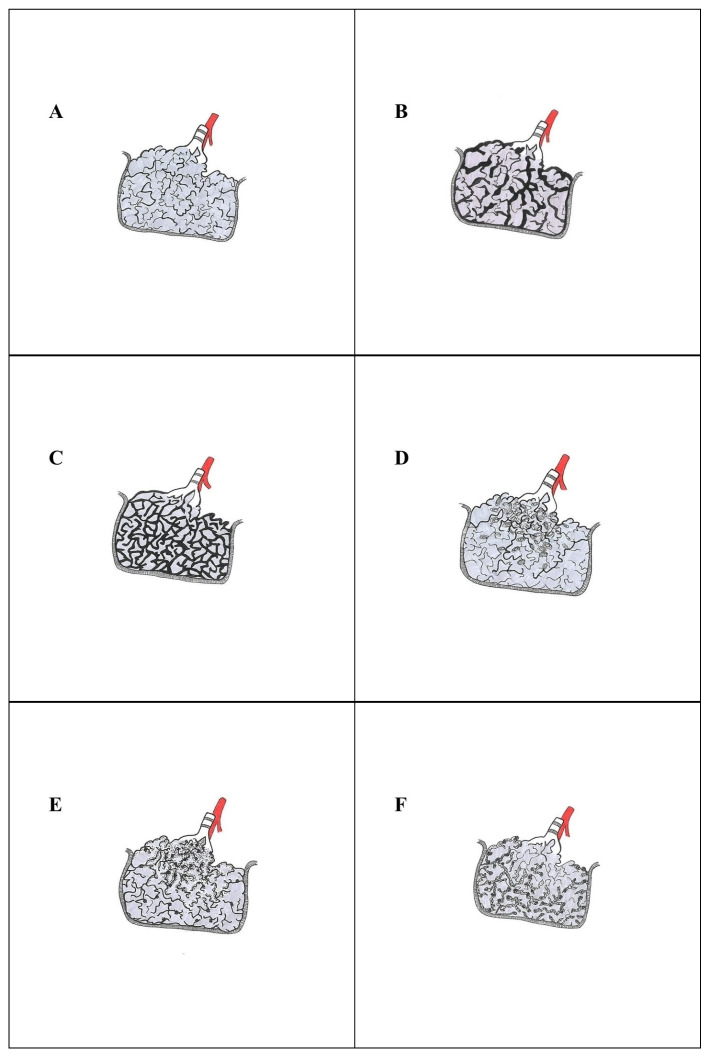
Schematic representation of normal SPL and key histopathological changes of IIPs. Normal secondary pulmonary lobule consisting of several primary lobules, supplied by the bronchiolovascular bundle (preterminal bronchiole and a branch of the pulmonary artery) and surrounded incompletely by fibrous interlobular septa (**A**). Heterogeneous appearance of UIP pattern with characteristic “patchwork” due to the patchy scarred fibrosis associated with a normal lung parenchyma (**B**). Temporal and geographical uniformity of NSIP pattern with diffuse and uniform interstitial fibrosing process characterizing the fibrotic type of disease (**C**). Patchy airspace filled by Masson bodies is the histopathological hallmark of the OP pattern (**D**). The basic element of the pathological DIP pattern is alveolar filling by pigmented macrophages, termed as “smokers’ macrophages” (**E**). Diffuse and dense cellular chronic interstitial infiltrate with a primarily alveolar septal distribution, with extensive infiltration and thickening of the alveolar septa, is the histological hallmark of the LIP pattern (**F**). (SPL, secondary pulmonary lobule; IIPs, Idiopathic Interstitial Pneumonias; UIP, Usual Interstitial Pneumonia; NSIP, Nonspecific Interstitial Pneumonia; OP, Organizing Pneumonia; DIP, Desquamative Interstitial Pneumonia; LIP, Lymphoid Interstitial Pneumonia).

## 3. Histological Diagnosis of Idiopathic Interstitial Pneumonias

### 3.1. Idiopathic Pulmonary Fibrosis—Usual Interstitial Pneumonia

Idiopathic Pulmonary Fibrosis (IPF) is the most common and aggressive form of IIP, characterised by chronic and progressive fibrosis associated with the unavoidable loss of lung function, progressive respiratory failure and high mortality [26]. The pathological hallmark of IPF is Usual Interstitial Pneumonia (UIP). Therefore, IPF is a multidisciplinary diagnosis, having the UIP histopathological pattern in the appropriate clinical and radiological setting [27]. Although diagnostic radiologic criteria have been developed in recent years that aim to avoid a biopsy, many patients still require histology for a correct diagnosis. The histopathological evaluation must follow strict diagnostic criteria [28]. Interstitial fibrosis associated with the architectural distortion of lung parenchyma with alveolar loss is the basic feature of UIP [22,29]. In particular, the extensive scar-like collagen deposition that characterizes the fibrosis seen in the UIP begins in the peripheral zones of SPL, where it is primarily seen (subpleural and paraseptal interstitium) [30,31]. The UIP pattern is heterogeneous in appearance at low magnification. Indeed, the characteristic “patchwork” of the UIP pattern observed at the scanning magnification is due to the patchy scarred fibrosis associated with a normal lung parenchyma (Figure 2B and Figure 3A). At the interface between dense fibrosis and normal lung fibroblasts, foci are observed (Figure 3B). These foci are small, oval-shaped or elongated stromal areas that are embedded in the myxoid stroma with a low collagen content, including spindle-shaped fibroblasts and myofibroblasts, layered parallel to the long axis of the alveolar septa. They occur within and along the interstitium, and their luminal surface is covered by flattened or hyperplastic type II pneumocytes or the bronchiolar epithelium, with possible associated squamous metaplasia. Fibroblast foci are required for the histopathological diagnosis of UIP, but are not specific to this pattern. Temporal and spatial heterogeneity are diagnostic of the UIP pattern [18,28,30,32,33,34,35,36]. Temporal heterogeneity is defined by a mixture of normal lung parenchyma, active and ongoing fibrosis (fibroblast foci) and chronic, inactive (scar-like) fibrosis (Figure 3A,B) [22]. Fibroblast foci represent a circumscribed area of reaction to acute lung injury, representing the first etiopathological event in UIP. Finally, the chronic progressive course with irreversible scarring and lung remodeling is explained by the numerous foci of microscopic acute lung injury that develop and recur throughout time [37,38,39]. On the other hand, spatial heterogeneity refers to the “patchwork” appearance of the UIP pattern, with a patchy distribution of scar-like fibrosis and architectural remodeling associated with areas of normal lung parenchyma. Generally, histological lesions of UIP are mainly observed in the subpleural and paraseptal interstitium with the relative absence of disease in the centrilobular region [40,41]. Architectural distortion, which is defined as the loss of normal alveolar structure and its replacement by scar-like fibrosis and/or honeycomb changes, observed both macroscopically and microscopically, is another histological finding that is required for the diagnosis of UIP (Figure 3C,D). The honeycomb change consists of reorganized, enlarged, and irregularly dilated cystic airspaces at least partially lined by ciliated (respiratory tract) epithelium (Figure 3C). These airspaces are also aggregated within dense collagen-type fibrosis and contain intraluminal mucin and inflammatory cells, such as neutrophils, macrophages, and lymphocytes. Scar-like fibrosis and honeycombing are hallmarks of end-stage lung disease, and they are associated with the irreversible loss of underlying lung structures (Figure 3D) [33,42,43,44,45,46]. In the context of dense fibrosis areas, mild chronic inflammation, prominent small blood vessels and, above all, smooth muscle hyperplasia, also with bundles of hyperplastic smooth muscle in cystic walls, should be detected. Elastic fibers from the original lung parenchyma discontinue or vanish in the lobule center, gathering instead in the perilobular zones [47]. Because of the altered lung parenchyma’s poor drainage, an acute or chronic inflammatory infiltrate might be seen in the honeycomb areas; however, in UIP, inflammatory infiltrates are usually mild. Occasionally multinucleated giant cells engulfed by cholesterol crystals are also observed, likely related to tissue breakdown. In addition, alveolar type II pneumocyte hyperplasia can be prominent in the areas of alveolar septal fibrosis and a blurry hyaline-like material, called “cytoplasmic or Mallory hyaline”, can be seen in the cytoplasm of the hyperplastic pneumocytes. Since most IPF patients smoke, some degree of persistent small airway remodeling is expected. Indeed, peribronchiolar metaplasia, associated with an increased number of intra-alveolar macrophages, representing respiratory bronchiolitis, is often seen. Finally, vascular changes such as intimal and medial hypertrophy may be present in UIP [48,49]. The key histopathological features of the UIP pattern are shown in Table 2 [50]. In this regard, the 2018 Official Clinical Practice Guideline of the American Thoracic Society (ATS)/European Respiratory Society (ERS)/Japanese Respiratory Society (JRS)/Latin American Thoracic Society (ALAT) recommends classifying patients with clinical–radiological suspicion of IPF into four histopathological categories based on morphological findings (Table 3). The histopathological criteria for UIP diagnosis reported in the guideline are:Dense fibrosis with architectural distortion;Predominant peripheral involvement of the secondary pulmonary lobule by fibrosis;Patchy involvement of lung parenchyma by fibrosis;The presence of fibroblast foci at the edges of the dense scars;The absence of features to suggest an alternative diagnosis.

Cases with all five features are classified as UIP. The other cases are instead classified as probable UIP or indeterminate UIP, based on the degree of morphological uncertainty, while the category of alternative diagnosis is for cases with highly specific histopathological features of other IIP patterns or indicating other ILDs. The pathological report strongly influences the final diagnosis in a multidisciplinary evaluation, as histopathological classification as “alternative diagnosis” excludes the possibility of IPF diagnosis. Therefore, the pathology report must be extremely accurate, and minor morphological findings suggesting an alternative diagnosis must be carefully evaluated [9,28]. Finally, a small percentage of IPF patients may experience an abrupt exacerbation of the disease, rapidly turning lethal. This scenario requires urgent medical attention [51,52]. The prototype of the histopathological pattern of acute exacerbation is a combination of the UIP pattern and superimposed Diffuse Alveolar Damage (DAD) with or without associated hyaline membranes [53,54,55,56]. However, although the acute exacerbation of IPF is a multidisciplinary diagnosis, pathologists have the option of suggesting it based on morphological findings [57,58].

**Figure 3 ijms-25-03618-f003:**
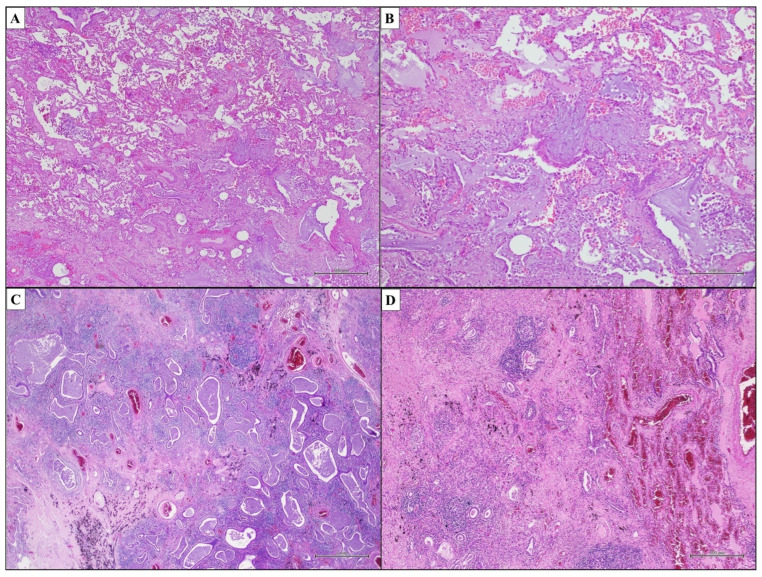
UIP pattern. Temporal and spatial heterogeneity diagnostic of UIP pattern ((**A**,**B**); hematoxylin and eosin, original magnifications ×500 and ×200). Temporal heterogeneity is defined by fibroblast foci localized at the interface between dense fibrosis and normal lung parenchyma ((**B**); hematoxylin and eosin, original magnifications ×200). Architectural distortion with normal alveolar structure replaced by scar-like fibrosis and/or honeycomb changes (**C**,**D**). Honeycomb change consists of reorganized, enlarged, and irregularly dilated cystic airspaces at least partially lined by ciliated (respiratory tract) epithelium ((**C**); hematoxylin and eosin, original magnifications ×1000). Normal lung parenchyma replaced by scar-like fibrosis define the end-stage lung disease ((**D**); hematoxylin and eosin, original magnifications ×500).

**Table 2 ijms-25-03618-t002:** Key histopathological features of UIP pattern (UIP, Usual Interstitial Pneumonia).

UIP Pattern
Dense interstitial fibrosis with lung architectural distortion (scar-like fibrosis and/or honeycomb change)
Patchy lung involvement (spatial heterogeneity)
Mainly subpleural and paraseptal involvement
Temporal heterogeneity of fibrosis (fibroblast foci and scar-like fibrosis)
Minimal interstitial inflammation

**Table 3 ijms-25-03618-t003:** Diagnostic categories and histopathological features of UIP given by Official American Thoracic Society/European Respiratory Society/Japanese Respiratory Society/Latin American Thoracic Association clinical practice guideline for the diagnosis of IPF (UIP, Usual Interstitial Pneumonia; IPF, Idiopathic Pulmonary Fibrosis; IIPs, Idiopathic Interstitial Pneumonias; LAM, Lymphangioleiomyomatosis) [28].

UIP	Probable UIP	Indeterminate for UIP	Alternative Diagnosis
Dense fibrosis with architectural distortion (i.e., destructive scarring and/or honeycombing) Predominant subpleural and/or paraseptal distribution of fibrosis Patchy involvement of lung parenchyma by fibrosis	Some histologic features from column 1 are present but to an extent that precludes a definite diagnosis of UIP/IPF And Absence of features to suggest an alternative diagnosis Or Honeycombing only	Fibrosis with or without architectural distortion, with features favoring either a pattern other than UIP or features favoring UIP secondary to another cause * Some histologic features from column 1, but with other features suggesting an alternative diagnosis **	Features of other histologic patterns of IIPs (e.g., absence of fibroblast foci or loose fibrosis) in all biopsies Histologic findings indicative of other diseases (e.g., hypersensitivity pneumonitis, Langerhans cell histiocytosis, sarcoidosis, LAM)
Fibroblast foci Absence of features to suggest an alternate diagnosis			

* Granulomas, hyaline membranes (other than when associated with the acute exacerbation of IPF, which may be the presenting manifestation in some patients), prominent airway-centered changes, areas of interstitial inflammation lacking associated fibrosis, marked chronic fibrous pleuritis, organizing pneumonia. Such features may not be overt or easily seen to the untrained eye and often need to be specifically sought. ** Features that should raise concerns about the likelihood of an alternative diagnosis include a cellular inflammatory infiltrate away from areas of honeycombing, prominent lymphoid hyperplasia including secondary germinal centers, and a distinctly bronchiolocentric distribution that could include extensive peribronchiolar metaplasia.

### 3.2. Idiopathic Nonspecific Interstitial Pneumonia—Nonspecific Interstitial Pneumonia 

The American Thoracic Society (ATS) recognized Idiopathic Nonspecific Interstitial Pneumonia (iNSIP) as an independent entity of IIP only in 2008. This discrete IIP is morphologically defined by the histopathological pattern of Nonspecific Interstitial Pneumonia (NSIP) [59]. Interestingly, the NSIP pattern could be found in a broad range of clinical settings, such as pulmonary diseases of known cause, like Connective Tissue Disease-Associated Interstitial Lung Disease (CTD-ILD) or Hypersensitivity Pneumonitis (HP), as well as in the setting of an IIPs [60,61,62,63,64,65,66,67,68,69]. The NSIP pattern is defined by an interstitial inflammatory and/or fibrosing process with diffuse and uniform distribution throughout the bioptic sample [59,70]. In detail, interstitial fibrosis, lymphocyte and/or plasma cell infiltrates, and temporal and geographical uniformity throughout the lesions are key elements of the pathologic pattern of NSIP (Figure 2C). A relatively diffuse and uniform expansion of alveolar septa caused by inflammation (Figure 4A–C) and/or fibrosis (Figure 4D) without the temporal and spatial heterogeneity that defines UIP should be noted. Despite the widespread nature of the NSIP pattern, the basic architecture of the lung is mostly preserved, but characterized by uniformly thickened alveolar septa and the lack of a completely normal lung (Figure 4A,B). This latter feature is determined on the basis of the temporal homogeneity of the disease [41,71,72]. According to the degree of inflammation and fibrosis, NSIP is subclassified into a cellular type and a fibrotic type. The primary feature of cellular NSIP (cNSIP) is an interstitial chronic inflammatory infiltration that mostly consists of mononuclear cells (lymphocytes and, to a lesser extent, plasma cells) without a remarkable rate of polymorphonuclears or histiocytes, and with no granulomas (Figure 4A–C). In many cases of cNSIP, reactive type II pneumocyte hyperplasia can be seen. Fibrotic NSIP (fNSIP) is predominantly characterized by diffuse and uniform interstitial fibrosis with associated alveolar architecture preservation (ordinary interstitial fibrosis) and lympho-plasmacytic infiltrate of a variable degree up to paucicellular or purely fibrotic form (Figure 2C and Figure 4D). Dense collagen deposition uniformly thickening the alveolar septa, peribronchiolar interstitium, interlobular and subpleural interstitium is characteristic of fibrosis in NSIP. However, fNSIP sometimes shows loose fibrosis, completely different from the scar-like fibrosis of the UIP, which is a mixture of edema, collagen fibers and a small number of fibroblasts. Moreover, the NSIP pattern lacks extensive fibroblast foci, smooth muscle hyperplasia and microscopic honeycombing [18,22,41,71,72,73,74]. Nonetheless, the differential diagnosis between fNSIP and UIP is extremely difficult because some cases of advanced fNSIP might exhibit “enlarged air-spaces” that resemble microscopic honeycombing [59,75]. Patchy intraluminal fibroblastic plugs resembling Organizing Pneumonia (OP) pattern can be seen, but they should be focal and should not extend over more than 20% of the region of disease [76]. The diagnosis of cNSIP rather than fNSIP has a significant clinical impact. Fibrosis is related with a worse prognosis. Therefore, a comment about the presence and the extension of fibrosis should always be made in the pathology reports of NSIP [77,78,79,80]. Key histopathological features of the NSIP pattern are shown in Table 4. 

### 3.3. Cryptogenic Organizing Pneumonia—Organizing Pneumonia

Organizing Pneumonia (OP) is currently recognized as a nonspecific lung injury pattern that has already been widely described under various names in the past [81,82,83,84]. OP can be secondary (Secondary Organizing Pneumonia, SOP), representing the histopathological manifestation of many clinical conditions such as CTD-ILD, drug reaction, HP, pulmonary infections and aspiration [85,86,87,88,89,90,91,92,93,94,95,96,97,98,99], or it may be not associated with a known cause or clinical condition, and appears to be idiopathic (Cryptogenic Organizing Pneumonia, COP) [27,100,101]. The histopathological hallmark of OP pattern is a patchy airspace filled by plugs of plump fibroblasts–myofibroblasts combined with variable numbers of admixed chronic inflammatory cells and embedded in a pale staining myxoid matrix of immature loose connective tissue with a polypoid shape, known as Masson bodies [22,101,102]. Tipically, Masson bodies are clustered in airspaces within and around small bronchioles and alveolar ducts, and they vary in shape according to the airspace size—round to elongated when localized in bronchioles, serpiginous and branching when localized in alveolar ducts and small and round when localized in alveolar spaces (Figure 2D) (Figure 5A,B). Furthermore, fibroblastic plugs can spread through Kohn pores from one alveolus to the next, creating the characteristic “butterfly pattern”. OP shows the typical patchy involvement of SPLs with relatively normal adjacent lung parenchyma. Alveolar ducts and alveolar spaces are mainly affected by bronchioles, which, when involved, can show lumen obstruction (hence the term bronchiolitis obliterans–organizing pneumonia in older literature). OP lesions characteristically show a peribronchial distribution and more rarely a paraspetal involvement. Immature connective tissue (young fibroblasia) of OP pattern appears to be the same age (lack of temporal heterogeneity). Giant cells and granulomas are usually absent, while foamy macrophages are observed in those alveoli spared by fibroplasia; mild to moderate interstitial lympho-plasma cellular inflammation, with mild thickening of the alveolar septa, can be detected. This mild interstitial pneumonia is usually confined to the OP areas or immediately adjacent parenchyma, and may be associated with mild reactive alveolar pneumocyte hyperplasia (Figure 5C). However, in the OP pattern, the basic alveolar architecture is relatively preserved, as highlighted by elastic fibers staining, with a lack of honeycomb change. It is essential to highlight that the mere presence of intra-alveolar fibroblastic plugs is not enough for the diagnosis of the OP pattern, as similar histological features have been shown to occur in several inflammatory lung diseases, including NSIP. Therefore, morphological features that may be expressions of other histopathological patterns and/or inflammatory process should be considered [18,27,72,101,103,104,105]. Key histopathological features of the OP pattern are shown in Table 5.

### 3.4. Acute Interstitial Pneumonia—Diffuse Alveolar Damage

Acute Interstitial Pneumonia (AIP) is an IIP characterized clinically by a rapid onset of respiratory failure associated with a pattern of Diffuse Alveolar Damage (DAD) upon histopathologic examination of the lung. This idiopathic disease was described by Katzenstein et al. in 1986 [106,107], but the same condition had probably previously been introduced by Hamman and Rich [108,109]. The DAD pattern includes the spectrum of pathologic changes that follow acute lung injury, and it is divided into three phases: exudative, organizing, and fibrotic [110,111]. The histologic hallmark of the exudative phase is the presence of hyaline membranes; this finding is associated with stromal and airspace edema, airspace epithelial denudation and the collapse of alveolar sacs, resulting in the simplification of the lung or widening of the alveolar duct [112] (Figure 6A,B). Afterwards, in the organizing phase, pathologists can observe the collapse of alveolar spaces, the thickening of the interstitium, squamous metaplasia and type II cell hyperplasia with reactive atypia as well, while hyaline membranes are extremely rare [113,114]. Finally, the fibrotic phase shows morphological overlap with fNSIP or the honeycomb lung of UIP. Key histopathological features of the DAD pattern are shown in Table 6.

### 3.5. Desquamative Interstitial Pneumonia—Desquamative Interstitial Pneumonia

Desquamative Interstitial Pneumonia (DIP) is an uncommon smoking-related IIP [27,115] that was first described by Liebow et al. in 1965 [116] and subsequently expanded in detail by Carrington et al. in 1978 [117]. Although DIP has been recognized as a distinct IIP, it shares extensive pathological similarities with Respiratory Bronchiolitis–Interstitial Lung Disease (RB-ILD), another smoking-related IIP, to the extent that there is debate over whether they are distinct diseases or a spectrum of the same condition [44,118,119]. However, patient outcomes differ enough to warrant separate classification and a commitment to a precise diagnosis, even though histological distinction may be challenging in some circumstances [120,121,122,123]. A basic element of the pathological DIP pattern is alveolar filling, and the histopathological hallmark is diffuse airspace filling by pigmented macrophages (Figure 2E) [22]. The term “smokers’ macrophages” refers to an airspace with pigmented macrophages because of the abundance of granules of yellow-brown to light brown color in the pale eosinophilic cytoplasm. Smokers’ macrophages accumulation affects the airspaces diffusely, and is not confined to peribronchiolar parenchyma as in RB-ILD. Scattered lymphocytes, eosinophils and multinucleated giant cells are often admixed with smokers’ macrophages within airspaces, but inflammatory cells are never numerous, and fibrinous exudates are absent. The lung architecture is preserved with alveolar septa only mildly thickened by uniform fibrosis and possible foci of emphysematous changes. Interestingly, the presence of lymphoid follicles, sometimes with germinal centers, in a peribronchiolar site is typical of the DIP pattern. Diffuse hyperplasia of type II pneumocytes is also observed. However, the diagnosis of DIP always requires a close correlation with clinical and radiological data, and a careful comparison with High-Resolution Computed Tomography (HRCT) is a strong recommendation [18,41,72,124,125,126,127,128]. Key histopathological features of the DIP pattern are shown in Table 7.

### 3.6. Respiratory Bronchiolitis-Interstitial Lung Disease—Respiratory Bronchiolitis

Respiratory Bronchiolitis–Interstitial Lung Disease (RB-ILD) is a rare IIP occurring almost exclusively in current or former heavy smokers [129,130]. The respiratory bronchiolitis (RB) pattern was first described in 1974 by Niewoehner et al. [131] as an incidental histological finding in the lungs of deceased cigarette smokers, and it is extremely common in this subset of patients. Afterwards, Myers et al. described a small series of patients with smoking history that showed the same histological features in surgical specimens associated with clinical symptoms, as well as clinical and imaging features, of ILD [132]. Most of the pathologic features of the RB pattern overlap with those of DIP, and often they cannot be accurately separated pathologically. The histologic hallmark of the RB pattern is the filling of only peribronchiolar airspaces (respiratory bronchioles, alveolar ducts, and immediately adjacent airspaces) by a large number of smokers’ macrophages with no significant interstitial inflammation or fibrosis [133]. So, the pathologic diagnosis should be RB [14], and the term RB-ILD should be used only in cases with clinico-radiological manifestations of ILD and after multidisciplinary discussion. 

### 3.7. Idiopathic Lymphoid Interstitial Pneumonia—Lymphoid Interstitial Pneumonia

Idiopathic Lymphoid Interstitial Pneumonia (iLIP) is a very uncommon ILD characterized by an extremely low prevalence, and placed in the category of rare IIPs in the updated 2013 IIP classification [27,134]. As well as other IIPs, iLIP also has its own histopathological appearance characterized by a Lymphoid Interstitial Pneumonia (LIP) pattern. The LIP pattern was first described by Liebow and Carrington in 1969 as a polymorphous lymphoid infiltrate that diffusely involves the alveolar septa interstitium [135]. The LIP pattern could be found in a broad range of clinical settings, including Legionella pneumonia and CTD-ILD (i.e., Sjögren’s Syndrome, Rheumatoid Arthritis, Systemic Lupus Erythematosus), or associated with other systemic diseases (i.e., Hashimoto’s Disease, Pernicious Anemia, Autoimmune Hemolytic Anemia, Primary Biliary Cirrhosis, Chronic Active Hepatitis, Myasthenia Gravis, Human Immunodeficiency Virus (HIV) Infection especially in children, Common Variable Immunodeficiency). Idiopathic cases are extremely rare [136,137,138,139,140,141,142,143,144,145,146,147]. A diffuse and dense cellular chronic interstitial infiltrate with a primarily alveolar septal distribution, composed of a mixture of mature plasma cells and polyclonal lymphocytes in varying proportions, is the histological hallmark of the LIP pattern (Figure 2F). The lymphocytes are predominantly T-cells with only scattered B-cells. The alveolar septa should be extensively infiltrated and severely thickened with compression and distortion of lung architecture, without or with minimal fibrosis (Figure 2F). Type II cell hyperplasia is often seen. Giant cells, also with cytoplasmic cholesterol clefts, small non-necrotizing poorly formed granulomas, and scattered lymphoid follicles, including follicles with germinal centers, are often present. A proteinaceous eosinophilic exudate is commonly observed within airspaces, and sometimes scattered intra-alveolar lymphocytes and macrophages are also evident. Focal features of the OP pattern (intra-alveolar organization) and DIP pattern (macrophage accumulation in air spaces) may also be present, but usually inconsistently. A considerable morphological feature of this pathological pattern is the diffuse distribution of the lesions without any specific localization, except for the lymphoid follicles located close to the pulmonary lymphatics [22,72,148,149,150,151,152,153,154]. The key histopathological features of LIP pattern are shown in Table 8. 

### 3.8. Idiopathic Pleuroparenchymal Fibroelastosis—Pleuroparenchymal Fibroelastosis

Idiopathic Pleuroparenchymal Fibroelastosis (iPPFE) is a rare IIP formally recognized and defined in the updated 2013 IIP classification [27,155]. The first description of this newly entity dates back to 1992 by Amitani et al. [156], and the term “Idiopathic Pleuroparenchymal Fibroelastosis” was coined eight years later by Frankel and colleagues [157]. Idiopathic Pleuroparenchymal Fibroelastosis appears as progressive pulmonary fibrosis mainly affecting the upper lobes. Its incidence and prevalence are unclear due to misdiagnoses, a lack of consensus regarding criteria for identification and ambiguities in its detection [158,159]. Prognosis seems to be poor, with a median survival of 2 to 5 years [160,161,162]. Although most Pleuroparenchymal Fibroelastosis (PPFE) cases are idiopathic [163,164,165,166], many potential initiating factors for PPFE have been reported, such as bone marrow and hematopoietic stem cell transplant, Graft-Versus-Host Disease (GVHD) [167,168,169], lung transplant [170,171], chemotherapy treatment [172], Autoimmune or Connective Tissue Disease (CTD-ILD) [173], Fibrotic Hypersensitivity Pneumonitis (fHP) [174], and occupational exposure to asbestos and aluminum [175]. Upper lobe/s elastofibrosis characterizes the PPFE pattern. This pathological feature resembles the well-known, nonspecific apical cap fibroelastosis of the lung, which is typically an incidental finding in lung biopsies. Although PPFE pattern elastofibrosis involves the lung more diffusely and is more widely distributed spatially, it is difficult or impossible to distinguish between the PPFE pattern and the pulmonary apical cap based solely on histopathology [176,177]. The basic elements of the pathologic pattern of PPFE are the temporally uniform elastofibrosis involving the pleura and subpleural lung parenchyma, predominantly in the upper lobe/s (Figure 7A,B). Pathologists must note a linear, or more often wedge-shaped, dense elastofibrotic scar without residual airspaces due to the complete filling of alveolar spaces by collagenous (apparently eosinophilic) and elastic (apparently basophilic) fibers. Traction bronchiolectasis and interstitial emphysema can be seen inside the elastofibrotic area, and a sharp interface, also with a small numbers of fibroblastic foci, between the scar and surrounding spared lung parenchyma is typical. As already mentioned, the overlying pleura are thickened by hyalinized fibrosis. There is no significant interstitial inflammatory infiltrate [18,27,41,178,179,180,181,182,183,184]. Watanabe et al. pointed out the primary role of histological evaluation in the diagnosis of iPPFE. The authors proposed criteria for the diagnosis of iPPFE, with definite diagnosis requiring histopathological study (Table 9) [185]. It is also necessary to highlight that several cases show PPFE’s histological features associated with pathological features of UIP or fNSIP in deeper parenchyma and/or in the lower lung. This finding could suggest that some cases may represent a combination of UIP and pulmonary apical cap. Therefore, the finding of the PPFE pattern on a biopsy sample should encourage the pathologists to carry out a careful examination of the underlying lung parenchyma for UIP or fNSIP, despite there being little consensus on how those cases should be classified [186]. Key histopathological features of the PPFE pattern are shown in Table 10. 

### 3.9. Unclassifiable Idiopathic Interstitial Pneumonia

Unclassifiable Idiopathic Interstitial Pneumonia (UCIP) is a category including all cases whose pathologic patterns do not fit into any specific type, those with a pattern indicating a new entity, or those with a mixture of more than one pattern [186,187]. Although rare, UCIP is not recommended as a histopathological diagnosis.

### 3.10. Rare Histological Patterns of Idiopathic Interstitial Pneumonia

Acute Fibrinous and Organizing Pneumonia (AFOP) and Bronchiolocentric Pattern of Interstitial Pneumonia are rare histological patterns currently lacking clinical and radiological data to define them as new disease entities. AFOP is characterized by the marked accumulation of intra-alveolar fibrin along with OP, with the specific morphological feature of fibrin balls inside the airspace (accumulation of fibrin should occupy more than 50% of airspaces within the lesion) [187]. Bronchiolocentric Pattern of Interstitial Pneumonia is a provisional condition described by different groups under various names (Bronchiolocentric Interstitial Pneumonia, Airway-Centered Interstitial Fibrosis and Peribronchiolar Metaplasia ILD), all sharing the airway-centered classification [188,189,190]. However, pathologists must always remember that our primary consideration when assessing airway-centered interstitial pneumonia is injury associated with inhalation, such as Hypersensitivity Pneumonitis or Aspiration Pneumonia, and not an idiopathic condition [191]. 

## 4. Combined Pulmonary Fibrosis and Emphysema: A Newly and Still Poorly Understood Clinico-Pathological Entity

Combined Pulmonary Fibrosis and Emphysema (CPFE) is a poorly understood clinical syndrome characterized by emphysema associated with fibrotic interstitial lung disease (fILD) [192,193]. The main risk factors are smoking and exposure to other aero-contaminants, which predispose genetically predisposed individuals to develop both fibrosis and emphysema. Most patients report a history of smoking, except those with CTD-ILD or fHP [174] who, on average, show less smoking exposure [194,195]. CPFE was defined based on clinical, physiologic, and HRCT features, while pathological aspects are still poorly considered due to the limited histological samples, represented by autopsy cases or explants [196,197]. However, emphysema associated with patterns of fILD can be seen in biopsy samples and, in this situation, pathologists must report these histopathological abnormalities. Evidence of emphysema is a necessary feature when looking to make the histopathological diagnosis of CPFE; pulmonary emphysema is defined as the abnormal and permanent enlargement of airspaces distal to the terminal bronchiole [198] and, based on the disease distribution within secondary pulmonary lobules, it can be classified into three major subtypes: centrilobular, panlobular and paraseptal emphysema [199]. Centrilobular emphysema is typically caused by cigarette smoking, and is often associated with paraseptal emphysema and other smoking-related abnormalities, such as RB, Smoking-Related Interstitial Fibrosis (SRIF), Langerhans Cell Histiocytosis (LCH) and DIP in CPFE patients. Patterns of fibrosis in CPFE are histologically heterogeneous [14]. A commonly encountered fibrosis is SIRF, a distinctive form of fibrosis linked to cigarette smoking [200]. SRIF is a common incidental finding in surgical lung specimens, and it is morphologically characterized by densely eosinophilic collagen deposited in the alveolar septa, with a predilection for subpleural and peribronchiolar parenchyma, the preservation of the overall lung architecture and little or no inflammatory infiltrate. The combination of SRIF and paraseptal emphysema explains the CPFE-specific histological finding of “thick-walled cystic lesions” [196,201]. However, SRIF, as well as LCH, without other patterns of concomitant fibrosis, are not diagnostic of CPFE to date, and, although CPFE is not exclusively a histopathological diagnosis, the combination of emphysema and a pattern of fibrosis other than SRIF or LCH, such as UIP, fNSIP or DIP, may suggest the diagnosis [202]. UIP fibrosis is the most commonly identified pattern of fibrosis in patients with CPFE, and its recognition is essential [203]. However, identifying the UIP pattern in the setting of concomitant emphysema can be extremely complex, and the typing of fibrosis is often impossible, with cases histologically classified as indeterminate for UIP [197]. 

## 5. Clinico-Radiological and Prognostic Features

The clinical presentation of IIPs widely ranges from acute/subacute to a chronic progressive onset of respiratory symptoms. Shortness of breath, exertional dyspnea possibly turning into persistent dyspnea during the latest stages of the disease, and diminished exercise tolerance are very common in chronic fibrosing IIPs, especially in IPF, iNSIP and smoking-related ILDs (i.e., DIP and RB-ILD). The dyspnea mechanisms are multifactorial, including increased lung elasticity, impaired gas diffusion, accessory muscles’ deconditioning and exhaustion, drug-induced myopathy, vascular endothelium injury, increased resistance in pulmonary circulation, as well chronic cor pulmonale [204] Chronic dry cough—a cough that lasts more than 8 weeks—is very common both in chronic and in subacute IIPs. The underlying mechanisms are far from being clarified. In patients with IPF, a significant alteration in sensory and/or inhibitory cough nerves or higher levels of neurotrophins causing a growth of sensory neurons have been described [205]. However, the cough significantly impacts the quality of life of ILDs patients, and possibly promotes fibrotic signaling, enhancing the mechanic stretch injury of the alveoli caused by wide and rapid endopleural pressure swings [206]. Subacute IIPs—in particular COP and some smoking-related ILDs—generally manifest inconstant degrees of cough and dyspnea for no more than 12 weeks. Acute IIPs are very rare; AIP leads to severe respiratory failure with several similarities to Acute Respiratory Distress Syndrome (ARDS). Spontaneous secondary pneumothorax and/or pneumomediastinum may represent the clinical onset of fibrosing IIPs, especially in IPF, iPPFE, and less frequently in COP, iNSIP or IIPs with cystic features (DIP, iLIP) [207]. Systemic symptoms may concur; unintentional weight loss and digital clubbing are more frequent in advanced chronic progressive fibrotic IIPs (IPF, iNSIP, iPPFE and DIP), while low-grade fever is more common in subacute and smoking-related ILDs. Uponn physical examination, late inspiratory crackles (velcro rales) are characteristic of many fibrotic IIPs. Pulmonary function tests usually reveal a restrictive pattern with reduced lung volumes and reduction in the diffusing lung capacity (DLCO) [208]. A mixed obstructive–restrictive pattern could be observed on lung function testing in patients with RB-ILD. More importantly, in fibrotic interstitial lung diseases, a decline in forced vital capacity (FVC) has been considered a surrogate for mortality, as 60–70% of deaths in IPF are associated with respiratory causes. Likewise, baseline low FVC and a reduction in FVC are linked with increased hospitalization and acute exacerbations in both IPF and non-IPF ILDs [209]. HRCT provides much information about radiological features, disease extension and progression over time. IIPs are associated with HRCT patterns that, in some cases, are exhaustive for the diagnosis (Figure 8). IPF is represented by a UIP pattern with a combination of honeycombing, reticulation abnormalities and traction bronchiectasis/bronchiolectasis simultaneously present on the HRCT scan. Other HRCT findings including volume loss, cranio-caudal gradient and poor/absent ground-glass opacities (GGOs) are coherent with UIP-pattern [210]. iNSIP is characterized by diffuse GGOs generally associated with reticulation and bronchiectasis involving mainly lower lobes. Subpleural sparing is often present, while honeycombing is rare (4.7%) [211]. In COP, the presence of sub-pleural or peribronchial consolidation areas with concurrent GGOs, band-like signs, perilobular patterns and reversed halo signs are commonly detected in thoracic HRCT scans [212]. In RB-ILD, the HRCT shows poorly formed centrolobular nodules with upper lobe predominance and the thickening of the bronchial walls. DIP usually presents a lower lobe-predominant ILD characterized by diffuse GGOs and signs of distortion in lung architecture such as traction bronchiectasis and lung cysts (resulting from dilated bronchioles or alveolar ducts); honeycombing is infrequent [213]. AIP shares similar HRCT scan features with ARDS, including a combination of mixed consolidative alveolar and GGOs with marked intralobular, and/or perilobular interstitial thickening resulting in a “crazy paving” pattern; unilateral or bilateral pleural effusion can be present [213]. iLIP is a very rare condition characterized by the middle and lower lobe predominance of ground glass opacities, thickened bronchovascular bundles, and some perilymphatic nodules; thin-walled lung cysts may concur in up to 68% of cases [213]. iPPFE is associated with irregular apex pleural thickening with upper lobe volume loss and significant architecture distortion; the anteroposterior flattening of the chest—platy thorax—may co-occur [214]. The clinical course and rate of progression of IIPs are extremely variable due to heterogeneity in disease-specific and patient-specific variables. The GAP index is a well-known, validated and simple mortality prediction model consisting of four parameters, including gender, age, FVC, and DLCO [215]. The GAP index allows for categorization into three stages with higher mortality rates for patients in more advanced stages. The GAP index has been validated in various patient populations with IPF and other ILDs [216]. Lately, an extended baseline clinical risk prediction model, the Distance–Oxygen–Gender–Age–Physiology (DO-GAP) index has been developed for patients with IPF. This model adds parameters of exercise capacity (6-min walk distance (6MWD) and exertional hypoxia) to the GAP index, and has demonstrated improved predictive performance for all-cause mortality [217]. HRCT as gold-standard non-invasive technique for the diagnostic evaluation of ILDs provides information about the radiological pattern and quantification of ILDs. However, recently, HRCT-based radiomics models including deep learning algorithms and artificial intelligence have been developed to prevent the risk of progression and the response to treatments in IPF and other fibrotic ILDs [218,219,220]. The early prediction of progression and identification of a functional or radiological signature predicting the clinical behavior can lead to prompt treatment for specific IIPs.

## 6. Lung Biopsy in Interstitial Lung Disease: A Multidisciplinary Choice

The choice of sampling method represents a critical point in the management of patients with ILDs [221,222]. The latest ATS/ERS/JRS/ALAT Clinical Practice Guideline highlights the value of multidisciplinary discussion in the choice of the best biopsy technique. Although surgical lung biopsy (SLB), especially video-assisted thoracic surgery (VATS), provides the best tissue samples for strict diagnosis, other biopsy techniques, such as bronchoscopic lung cryobiopsy (BLC) and transbronchial biopsy (TBB), may be effectively performed [223]. SLB is considered the gold standard diagnostic procedure to establish the histopathologic pattern of a specific type of ILD, allowing a definitive diagnosis in almost 90% of clinical–radiologically suspected IPF [224,225]. SLB is recommended for cases of newly detected ILD of apparently unknown cause, clinically suspect for IPF and having an HRCT pattern of probable UIP, indeterminate for UIP, or an alternative diagnosis [226]. Although large samples containing peripheral structures of the secondary pulmonary lobule are obtained via this bioptic approach, SLB is burdened by some risks and complications, such as a significant mortality at 90 days, mainly for the acute exacerbation of IPF [227], and it is not advisable in subgroups of patients (older subjects, patients with significant comorbidities) [228,229,230]. Transbronchial lung cryobiopsy (TBLC) has been proposed as an equally informative and less invasive alternative to SLB [231,232,233,234]. This technique is based on the Joule–Thompson effect, and it works with a cryoprobe. TBLC has shown several benefits compared to other biopsy approaches. It provides larger lung tissue samples, up to 64 mm^2^ in size, containing some peripheral structures of the secondary pulmonary lobules, with fewer crush artefacts compared with TBB. A good cryobiopsy should be at least 5 mm on the longest axis and must obviously contain the radiologically suspicious areas [235]. This approach is safer than operative biopsy, and the occurrences of the main side effects, i.e., pneumothorax and bleeding, are comparable to regular forceps biopsies. Moreover, although the observation of the peripheral zone inside the secondary pulmonary lobule is partially limited, lacking the subpleural area, TBLC shows a high diagnostic yield of nearly 80% with an excellent safety profile compared with SLB [232,236,237,238,239,240,241]. All data demonstrate a moderate diagnostic accordance between TBLC and SLB in the setting of the multidisciplinary diagnosis of diffuse lung disease, mainly when performed in experienced centers [242,243,244]. Conversely, the diagnostic values of TBB and bronchoalveolar lavage (BAL) are limited for IIPs, with TBB findings often nondiagnostic [245,246], while the cellular analysis of BAL may help in defined clinical settings, such as suspected HP, eosinophilic pneumonia, sarcoidosis and infections [247,248,249,250]. TBLC is commonly indicated for diffuse parenchymal lung disease when radiological and clinical data do not yield a definitive diagnosis, as it provides enough histopathological information to reach a multidisciplinary diagnosis in patients with diffuse lung disease, with the same diagnostic value but lower risk compared to SLB. Moreover, TBLC allows more patients with diffuse lung disease to benefit from histological evaluation, such as patients in whom the severity of the disease and/or potential complications would preclude SLB or patients with mild/early disease, allowing early diagnosis in the setting of interstitial lung anomalies on the HRCT [235,251]. 

## 7. Molecular Testing as a Useful Partner of Morphology: The Beginning of a New Era

Idiopathic pulmonary fibrosis is the most common idiopathic progressive fibrosing disease; it shows poor prognosis and represents a diagnostic challenge [28]. Recent studies have highlighted misdiagnosis or delay in IPF diagnosis, often resulting in harmful or delayed therapies [252,253,254,255,256,257,258,259,260]. Although histopathology plays a major role in multidisciplinary discussions to reach a diagnosis consensus, unfortunately, not all patients with suspected IPF can tolerate surgery [227,261,262,263]. The Envisa Genomic Classifier (EGC) is a laboratory test developed to identify a gene expression signature concordant with the UIP pathology in minimally invasive TBB [264,265,266]. The EGC uses a next-generation RNA-sequencing assay to enrich and amplify exonic transcripts from 15 ng total RNA, extracted and pooled from 3–5 TBBx per patient. The classifier consists of 190 genes to identify a molecular UIP pattern. Expression count data are input into a locked and validated machine learning algorithm for classification, developed for high rule-in performance characteristics, trained using a cohort of 90 ILD patients [265,266,267]. Some validation studies [265,268,269,270] have demonstrated the utility of the test in distinguishing UIP patterns from non-UIP patterns, and its impact on multidisciplinary teams’ clinical decision-making in the diagnosis of IPF (Table 11). Four studies [265,268,269,270], with a total cohort of 200 patients, assessed the diagnostic value of the test; their individual data show a sensitivity ranging from 59% to 80% and a specificity ranging from 78% to 100%. Raghu et al. [267], Richeldi et al. [268] and Kheir et al. [269] used a composite of clinical, radiographic, and histopathologic data as a reference standard, while for Pankratz et al. [264], the reference standard was histopathology alone. When aggregated by meta-analysis, EGC identified the UIP pattern with a sensitivity and specificity of 68% (CI, 55–73%) and 92% (CI, 81–95%), respectively, in patients with ILD [271]. The agreement of the categorization and the diagnostic confidence obtained with and without the use of genomic classifier data were also evaluated. The reported agreements were of 86% (95% CI, 78–92%) [268] and of 88% (95% CI, 67–97%) [270] between IPF and non-IPF clinical diagnoses made with molecular tests and histopathology, and with molecular tests and the multidisciplinary approach, respectively. The EGC was also demonstrated to improve diagnostic confidence, with increases from 56% to 89% and 43% to 93% when considered in the context of multidisciplinary evaluation [268,270]. In this regard, Lasky et al. [272] studied the impact of Envisa applied to TBB on physicians’ clinical decision-making in the diagnosis and management of IPF. Their results show that the number of IPF diagnoses increased from 30% to 69% with genomic classifier testing. Moreover, a high confidence (≥90%) in ILD diagnoses was more common, and the recommendation of antifibrotic treatment increased from 10% to 46.4% with EGC [272]. Thus, considering its high specificity (≥86%) in predicting UIP pattern, EGC provides important diagnostic information in multidisciplinary meetings, and it is particularly useful in settings without immediate access to an ILD expert center, increasing the diagnostic confidence for patients with clinical probable UIP and reducing the need for additional and more invasive sampling by SLB [273]. On the other hand, the use of genomic tests in clinical practice is still premature due to its low sensitivity, with many cases still requiring SLB, given the earliness of the data. However, although the 2022 ATS/ERS/JRS/ALAT IPF clinical practice guideline update has made no recommendation for or against the use of genomic classifier testing in fibrotic ILD diagnosis, the clinical utility of EGC is promising, and it seems to be useful when used to ascertain the diagnosis of IPF and ensure excellent therapeutic management, especially for patients not eligible for SLB. At the same time, the application of computer-based deep learning algorithms in the assessment of ILD histopathology specimens represents another interesting diagnostic perspective [273,274]. Mäkelä et al. [275] first developed and tested the ability of a semi-supervised deep learning algorithm to identify and quantify specific ILD features, such as fibroblastic foci, interstitial mononuclear inflammation, and intra-alveolar macrophages, in lung tissue samples. This study shows that automated image analysis could provide great assistance in the identification and quantification of ILD’s histopathological features, and artificial intelligence could be a novel tool for pathologists to use in the histological diagnosis of IPF and other interstitial lung disorders in the future. Finally, other molecular biomarkers, i.e., alterations in telomere-related genes [276], serum and tissue, have been studied as potential diagnostic tools or prognostic and therapeutic factors. Although potentially useful resources in the diagnostic–therapeutic management of patients with ILDs, none of them have been validated in large clinical trials [277]. 

## 8. Multidisciplinary Team: Sharing and Discussion

The multidisciplinary approach is essential for the management of patients with ILDs, especially fibrotic ILDs, and the use of a multidisciplinary team (MDT) is the gold standard for these diagnoses [14,28]. Several Consensus Statements and Guidelines recommend that MDT should, as a minimum, include a pulmonologist, a thoracic radiologist and a thoracic pathologist in order to integrate different clinical, radiological and histopathological data in order to establish a final diagnosis [28,50]. The multidisciplinary approach has widely demonstrated that a comprehensive assessment of clinical, radiological and pathological features improves the diagnostic agreement and clinical management of patients [278]. However, it should be considered that other physicians are notable contributors in patient management; for example, rheumatologists and transplant surgeons. ILDs, particularly fibrotic types, are often classified based on diagnostic confidence as confident diagnosis, provisional diagnosis and unclassifiable, and the multidisciplinary approach has the task of evaluating the level of confidence in diagnosis, in order to propose a first-choice diagnosis and to identify alternative diagnoses [279]. Focusing on the pathologist, its role is to interpret the histopathological features of lung biopsy and contribute to diagnosis. Guidelines recommend performing biopsy only if the clinical–radiological evidence is not diagnostic, and MDT discussion is essential in this setting. Based on clinical suspicion, an MDT discussion may help in choosing the best biopsy method and in determining the most appropriate area for tissue sampling, so that the sample is histologically adequate to provide diagnostic information to pathologist. Particularly in institutions lacking expertise in TBLC, MDT discussion can guide the selection of patients for SLB or, in selected case, for TBB, which may be useful in the assessment of processes with a lymphangitic distribution, such as sarcoidosis or silicosis, or bronchiolocentric distribution, such as HP or OP [280]. Furthermore, pathologists can explain the histological report and eventually gene expression profile data in MTD discussion, in order to correlate their findings with those of other specialists and so establish the correct diagnosis [281]. Finally, there is not always diagnostic agreement between physicians, especially between radiologists and pathologists. This finding applies more to the non-UIP than to the UIP pattern. Travis et al. [59] observed that among 105 biopsy-proven NSIP patients, 21 were disconfirmed after merging radiological data. Similar data have been presented by Flaherty et al. [282]. In their study, HRCT findings of ground-glass opacity and reticular abnormality were not accurate predictors of histological NSIP, and in their series, most of the patients with typical HRCT findings of NSIP had a histological diagnosis of UIP upon surgical lung biopsy (26/44, 59%). Moreover, they observed that the HRCT features of UIP were seen in only a few of the patients with a histological diagnosis of UIP (27/73, 37%), and heterogeneous data have been reported regarding the agreement between the histological and radiological diagnosis of UIP [283,284,285]. This suboptimal concordance is also reported for other IIPs patterns [286]. In these circumstances, the histological examination often showed pathological features of specific patterns, and the cases were re-evaluated through multidisciplinary discussion, concluding on a different diagnosis. However, there is an adequate agreement on UIP pattern diagnosis, and this is extremely relevant considering that the differential diagnosis between IPF and some other fibrotic ILDs is essential because of the worse prognosis [287].

## 9. Conclusions

IIPs are uncommon lung disorders with often difficult-to-interpret histological features. Many idiopathic and non-idiopathic ILDs share the same histological features. Multidisciplinary evaluation is essential in this clinical setting. The pathologist’s aim is to define the histopathological pattern of disease; a definite diagnosis of a specific IIP can be achieved only after multi-disciplinary discussion, upon correlation with clinical–radiological findings, allowing optimal clinical–therapeutic management of the patient.

## Figures and Tables

**Figure 1 ijms-25-03618-f001:**
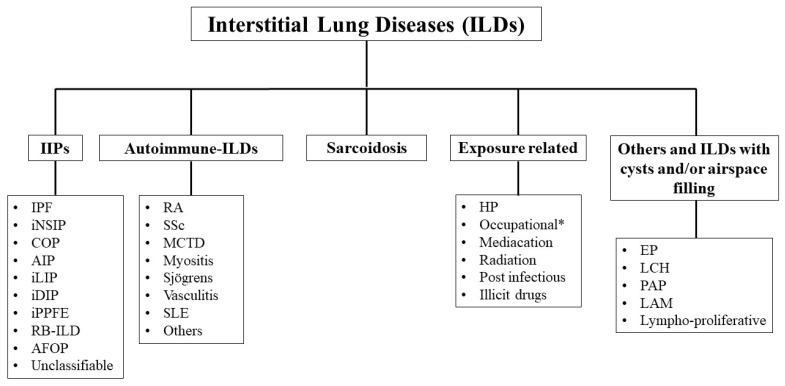
Current classification of Interstitial Lung Diseases (ILDs) [9]. Idiopathic Interstitial Pneumonias (IIPs), Idiopathic Pulmonary Fibrosis (IPF), Idiopathic Nonspecific Interstitial Pneumonia (iNSIP), Cryptogenic Organizing Pneumonia (COP), Acute Interstitial Pneumonia (AIP), Idiopathic Lymphoid Interstitial Pneumonia (iLIP), Idiopathic Desquamative Interstitial Pneumonia (iDIP), Idiopathic Pleuroparenchymal Fibroelastosis (iPPFE), Respiratory Bronchiolitis–Interstitial Lung Disease (RB-ILD), Acute Fibrinous and Organizing Pneumonia (AFOP), Rheumatoid Arthritis (RA), Systemic Sclerosis (SSc), Mixed Connective Tissue Disease (MCTD), Systemic Lupus Erythematosus (SLE), Hypersensitivity Pneumonitis (HP), Eosinophilic Pneumonia (EP), Langerhans Cell Histiocytosis (LCH), Pulmonary Alveolar Proteinosis (PAP), Lymphangioleiomyomatosis (LAM). * Occupational: asbestosis, silicosis, coal miner, berylliosis, others.

**Figure 4 ijms-25-03618-f004:**
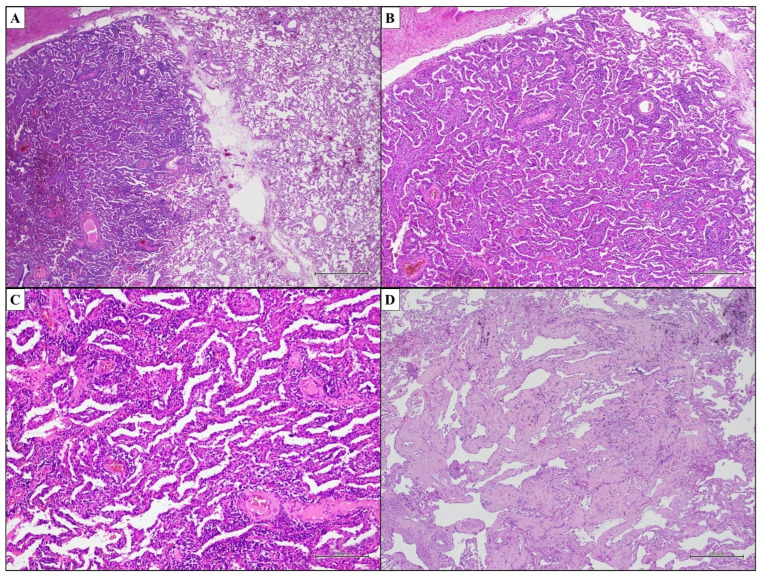
NSIP pattern. Temporal and spatial uniformity of NSIP pattern ((**A**,**B**); hematoxylin and eosin, original magnifications ×1000 and ×500). Diffuse and uniform interstitial chronic inflammatory infiltrate of cNSIP ((**C**); hematoxylin and eosin, original magnifications ×200). Diffuse and uniform interstitial fibrosis with associated lympho-plasmacytic infiltrate of variable degrees of fNSIP ((**D**); hematoxylin and eosin, original magnifications ×500). (cNSIP, cellular Nonspecific Interstitial Pneumonia; fNSIP, fibrotic Nonspecific Interstitial Pneumonia).

**Figure 5 ijms-25-03618-f005:**
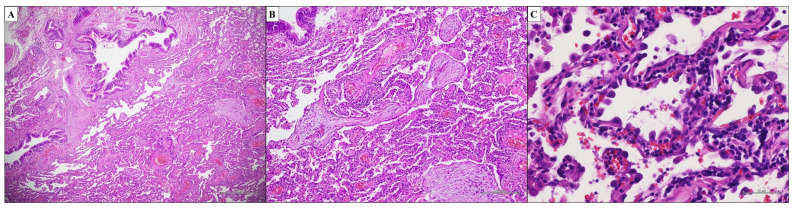
OP pattern. Patchy airspace filled with Masson bodies. Masson bodies are clustered in airspaces within and around small bronchioles and alveolar ducts, and they vary in shape according to the airspace’s size—round to elongated when localized in bronchioles, serpiginous and branching when localized in alveolar ducts and small and round when localized in alveolar spaces ((**A**,**B**); hematoxylin and eosin, original magnifications ×500 and ×200). Reactive alveolar pneumocyte hyperplasia in OP ((**C**); hematoxylin and eosin, original magnifications ×50).

**Figure 6 ijms-25-03618-f006:**
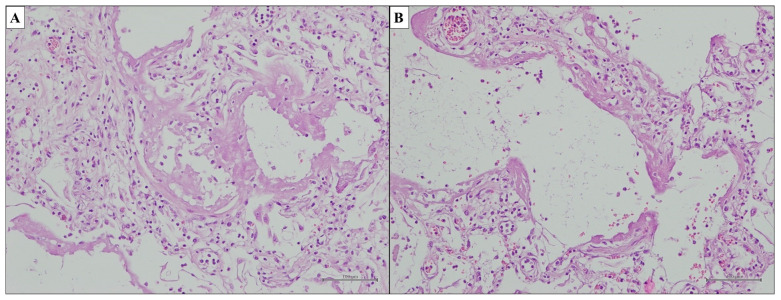
DAD pattern. Hyaline membranes are the histologic hallmark of the DAD exudative phase; they are associated with airspace epithelial denudation ((**A**,**B**); hematoxylin and eosin, original magnifications ×100 and ×100).

**Figure 7 ijms-25-03618-f007:**
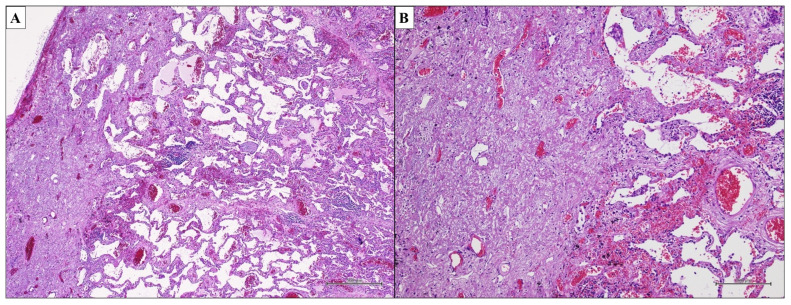
PPFE pattern. The basic elements of the PPFE pattern are temporally uniform elastofibrosis involving the pleura and subpleural lung parenchyma ((**A**,**B**); hematoxylin and eosin, original magnifications ×500 and ×200).

**Figure 8 ijms-25-03618-f008:**
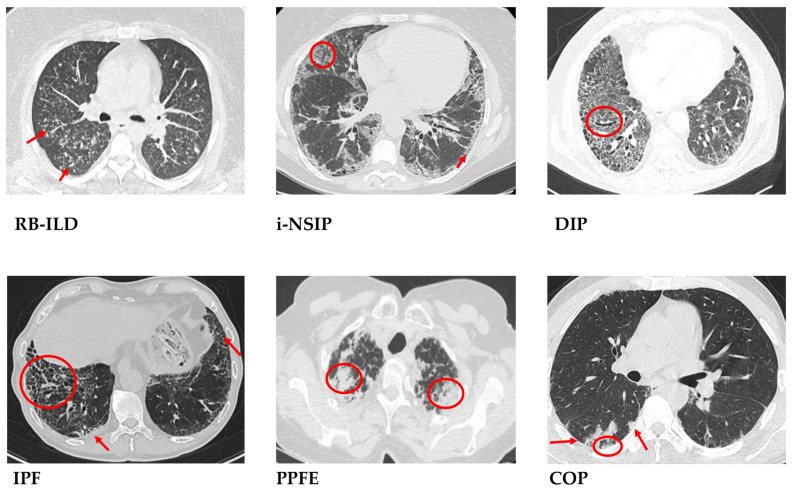
Thoracic computed tomography of the main Idiopathic Interstitial Pneumonias (IIPs). Respiratory Bronchiolitis–Interstitial Lung Disease (RB-ILD) is characterized by poorly formed centrolobular nodules (arrows); Idiopathic Nonspecific Interstitial Pneumonia (iNSIP) is characterized by patchy ground-glass opacities (GGOs) possibly with traction bronchiectasis (circle) and reticulations (arrow); Desquamative Interstitial Pneumonia (DIP) is characterized by homogeneous fibrotic GGOs with traction bronchiectasis (circle) possibly associated with cysts; Idiopathic Pulmonary Fibrosis (IPF) is associated with patchy areas of honeycombing with traction bronchiectasis (circle) with reticulation (arrow); Pleuroparenchymal Fibroelastosis (PPFE) is characterized by bilateral irregular apex thickening (circles); Cryptogenetic Organizing Pneumonia (COP) is associated with peribronchial or peripheral consolidative areas (arrows) possibly associated with perilobular pattern or reverse halo sign (circle).

**Table 1 ijms-25-03618-t001:** American Thoracic Society/European Respiratory Society International Multidisciplinary classification of the IIPs (IIPs, Idiopathic Interstitial Pneumonias).

Category	Clinical–Radiological–Pathologic Diagnosis	Pathological Patterns
Major IIPs		
Chronic Fibrosing Interstitial Pneumonia	Idiopathic Pulmonary Fibrosis	Usual interstitial pneumonia
	Idiopathic Nonspecific Interstitial Pneumonia	Nonspecific interstitial pneumonia
Acute/Subacute Interstitial Pneumonia	Cryptogenic Organizing Pneumonia	Organizing Pneumonia
	Acute Interstitial Pneumonia	Diffuse Alveolar Damage
Smoking-Related Interstitial Pneumonia	Respiratory Bronchiolitis–Interstitial Lung Disease	Respiratory Bronchiolitis
	Desquamative Interstitial Pneumonia	Desquamative Interstitial Pneumonia
Rare IIPs		
	Idiopathic Lymphoid Interstitial Pneumonia	Lymphocytic Interstitial Pneumonia
	Idiopathic Pleuroparenchymal Fibroelastosis	Pleuroparenchymal Fibroelastosis
	Unclassifiable Idiopathic Interstitial Pneumonia	Variable
Rare Histologic Patterns		
	Acute Fibrinous and Organizing Pneumonia	
	Bronchiolocentric Patterns of Interstitial Pneuomnia	

**Table 4 ijms-25-03618-t004:** Key histopathological features of NSIP pattern (NSIP, Nonspecific Interstitial Pneumonia; cNSIP, cellular Nonspecific Interstitial Pneumonia; fNSIP, fibrotic Nonspecific Interstitial Pneumonia; OP, Organizing Pneumonia).

cNSIP Pattern	fNSIP Pattern
Uniform mild-to-moderate interstitial lymphoplasmacellular inflammation	Dense or loose interstitial fibrosis with uniform appearance (temporal homogeneity)
Reactive Type 2 pneumocyte hyperplasia in areas of inflammation	Basic architecture of the lung preserved (also demonstrated by elastic fibers staining)
Basic architecture of the lung preserved (also demonstrated by elastic fibers staining)	Variable degree of interstitial chronic inflammation
OP pattern not the prominent, if present (<20% of biopsy specimen)	Microscopic honeycombing often inconspicuous, if present
	Lack of temporal heterogeneity (scar-like fibrosis—fibroblastic foci—normal lung)
For both types Lack of Diffuse alveolar damage pattern Eosinophilic infiltrate: inconspicuous or absent Lack of granulomas Lack of viral inclusions and organisms on special stains Lack of dominant airway disease such as extensive peribronchiolar metaplasia	

**Table 5 ijms-25-03618-t005:** Key histopathological features of OP pattern (OP, Organizing Pneumonia).

OP Pattern
Airspace filled by Masson bodies
Basic architecture of the lung preserved (also demonstrated by elastic fibers staining)
Patchy distribution
Temporal uniformity of fibrosis (young fibroplasia)
Mild to moderate interstitial chronic inflammation

**Table 6 ijms-25-03618-t006:** Key histopathological features of DAD pattern (DAD, Diffuse Alveolar Damage).

DAD Pattern
Focal or diffuse hyaline membranes
Diffuse involvement of lung parenchyma
Temporal uniformity
Alveolar septal thickening due to organisation
Diffuse type II pneumocytes hyperplasia also with reactive atypia

**Table 7 ijms-25-03618-t007:** Key histopathological features of DIP pattern (DIP, Desquamative Interstitial Pneumonia).

DIP Pattern
Prominent and diffuse airspace filed by smokers’ macrophages
Uniform involvement of lung parenchyma
Basic architecture of the lung preserved (also demonstrated by elastic fibers staining)
Evidence of multiple lymphoid follicles, also with germinal centers
Lack of prominent interstitial fibrosis and inflammatory infiltrate
Diffuse type II pneumocytes hyperplasia

**Table 8 ijms-25-03618-t008:** Key histopathological features of LIP pattern (LIP, Lymphoid Interstitial Pneumonia).

LIP Pattern
Diffuse interstitial lymphoplasmacellular infiltrate
Predominantly alveolar septal distribution
Marked widening of alveolar septa with compression and distortion of alveolar spaces
Evidence of lymphoid follicles, also with germinal centers, close to the pulmonary lymphatics

**Table 9 ijms-25-03618-t009:** Criteria for the diagnosis of definite iPPFE proposed by Watanabe et al. (iPPFE, Idiopathic Pleuroparenchymal Fibroelastosis; HRCT, High-Resolution Computed Tomography; PPFE, Pleuroparenchymal Fibroelastosis) [185].

Definite iPPFE		
Multiple subpleural foci of airspace consolidation with traction bronchiectasis located predominantly in the bilateral upper lobes on HRCT scans	Subpleural zonal or wedge-shaped dense fibrosis consisting of collapsed alveoli and collagen-filled alveoli with septal elastosis, with or without collagenous thickening of visceral pleura in surgical lung biopsy specimens	Exclusion of other diseases with known causes or conditions showing radiological and/or histological PPFE patterns such as chronic hypersensitivity pneumonia, connective tissue diseases, occupational diseases, hematopoietic stem cell or lung transplantation-related lung diseases
Definite iPPFE is diagnosed when all three criteria are met. If the lower lobes are affected by fibrosis, multidisciplinary discussion is necessary for the final diagnosis.		

**Table 10 ijms-25-03618-t010:** Key histopathological features of PPFE pattern (PPFE, Pleuroparenchymal Fibroelastosis).

PPFE Pattern
Prominent and homogenous (lack of temporal heterogeneity) subpleural elastofibrosis of the upper lobe/s
Hyalinized fibrosis of overlying pleura
Traction bronchiolectasis and interstitial emphysema
At most, small numbers of fibroblastic foci at the interface
At most, mild lymphoplasmocytic infiltrates

**Table 11 ijms-25-03618-t011:** Validation studies of Envisa Genomic Classifier (EGC) in distinguishing the UIP pattern from the non-UIP pattern. (CI, confidence interval; PPV, positive predictive value; NPV, negative predictive value; ILD, interstitial lung disease; MD, multidisciplinary discussion).

Reference	No of Patients	Population	Reference Standard	Sensitivity [95% Cl]	Specificity [95% Cl]	PPV	NPV
Pankratz et al. [264]	31	Suspected ILD	Histopathology	0.59 [0.33, 0.82]	1.00 [0.66, 1.00]	100%	56.2%
Kheir et al. [269]	24	Suspected ILD	MD	0.80 [0.52, 0.96]	0.78 [0.40, 0.97]	85%	70%
Raghu et al. [267]	49	Suspected ILD	MD	0.70 [0.47, 0.87]	0.88 [0.70, 0.98]	84.2%	76.6%
Richeldi et al. [268]	96	Suspected ILD	MD	0.60 [0.47, 0.73]	0.92 [0.79, 0.98]	92%	60.3%

## Data Availability

No new data were created.

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
