# Peer review of "Multidisciplinary Approach to the Diagnosis of Idiopathic Interstitial Pneumonias: Focus on the Pathologist’s Key Role"

_ijms, 2024, doi:10.3390/ijms25073618_

Round 1

Reviewer 1 Report

Comments and Suggestions for Authors

Good work which contributes to a better description and classification of chronic diffuse interstitial pneumonia, however, it is interesting to highlight the trend towards the reduction of lung biopsies in favor of imaging and probably Molecular testing shortly 

Author Response

Reviewer 1 Good work which contributes to a better description and classification of chronic diffuse interstitial pneumonia, however, it is interesting to highlight the trend towards the reduction of lung biopsies in favor of imaging and probably Molecular testing shortly 

Response to reviewer: Dear Reviewer, thank you very much for taking the time to review this manuscript. Your comment is extremely relevant and we have explained better the usefulness of the molecular testing especially in the multidisciplinary discussion. We have also highlighted how this approach is highly specific and provides important diagnostic information in multidisciplinary meetings, increasing the diagnostic confidence for patients with clinical probable UIP and reducing the need of additional and more invasive sampling. (page number: 21, paragraph: 7, lines 690 to 712)

Reviewer 2 Report

Comments and Suggestions for Authors

Comment

High resolution CT (HRCT) scanning has contributed significantly to the evaluation of patients with interstitial lung disease (ILD) and is particularly useful in the diagnosis of idiopathic pulmonary fibrosis (IPF). The characteristic radiographic features of the idiopathic interstitial pneumonias (IIP) on HRCT scans. The HRCT scanning can provide a confident, highly specific diagnosis of IPF in many patients with diffuse lung disease. The role of surgical lung biopsy is discussed in the diagnosis of cases when a definite HRCT diagnosis cannot be made.

1 In this draft , the term HRCT is precise ,not the Computed Tomography(CT)

2 The figure 8 , I suggest , the authors use the arrow or circle to indicate the region of the

A) UIP-pattern with a combination of honeycombing, reticulation and traction bronchiectasis/bronchiolectasis

B) UIP-pattern

C) iNSIP is characterised - GGOs ,reticulation and bronchiectasis .

D) In COP, the presence of sub-pleural or peribronchial consolidation areas with con current GGOs, band-like signs, perilobular pattern and reversed halo signs.

E) In RD-ILD, the CT shows poorly formed centrolobular nodules with upper lobe predominance and thickening of the bronchial walls.

F) DIP usually presents a lower lobe-predominant ILD characterised by diffuse GGOs and signs of distortion in lung architecture as traction bronchiectasis and lung cysts (resulting from dilated bronchioles or alveolar ducts); honeycombing is infrequent. G)AIP shares similarities with ARDS, including a combination of mixed consolidative alveolar and GGOs with marked intralobular and/or perilobular interstitial thickening resulting in ‘crazy paving’ pattern; unilateral or bilateral pleural effusion can be present .

H)iLIP is a very middle and lower lobes predominance of ground glass opacities, thickened of bronchovascular bundles, some perilymphatic nodules; thin walled lung cysts

I)iPPFE is associated with irregular apex pleural thickening with upper lobe volume loss and significant architecture distortion; anteroposterior flattening of the chest platy thorax may coexist

Where is reverse halo sign ?

Author Response

Reviewer 2 High resolution CT (HRCT) scanning has contributed significantly to the evaluation of patients with interstitial lung disease (ILD) and is particularly useful in the diagnosis of idiopathic pulmonary fibrosis (IPF). The characteristic radiographic features of the idiopathic interstitial pneumonias (IIP) on HRCT scans. The HRCT scanning can provide a confident, highly specific diagnosis of IPF in many patients with diffuse lung disease. The role of surgical lung biopsy is discussed in the diagnosis of cases when a definite HRCT diagnosis cannot be made.

Comment to reviewer: Dear Reviewer, thank you very much for taking the time to review this manuscript and for your interesting suggestions with which we improved our work.

Comment 1 In this draft, the term HRCT is precise, not the Computed Tomography(CT)

Response 1: We have changed the term CT with HRCT in in each paragraph of the text.

Comment 2 The figure 8, I suggest, the authors use the arrow or circle to indicate the region of the

  1. A) UIP-pattern with a combination ofhoneycombing, reticulationand traction bronchiectasis/bronchiolectasis
  2. B) UIP-pattern
  3. C) iNSIP is characterised - GGOs, reticulation and bronchiectasis .
  4. D) In COP, the presence of sub-pleural or peribronchial consolidationareas with con current GGOs, band-like signs, perilobular pattern and reversed halo signs.
  5. E) In RD-ILD, the CT shows poorly formedcentrolobular nodules with upper lobe predominance and thickening of the bronchial walls.
  6. F) DIP usually presents a lower lobe-predominant ILD characterised by diffuse GGOs and signs of distortion in lung architecture as traction bronchiectasis and lung cysts (resulting from dilated bronchioles or alveolar ducts); honeycombing is infrequent. G) AIP shares similarities with ARDS, including a combination of mixed consolidative alveolar and GGOswith marked intralobular and/or perilobular interstitial thickening resulting in ‘crazy paving’ pattern; unilateral or bilateral pleural effusion can be present.
  7. H) iLIP is a very middle and lower lobes predominance of ground glass opacities, thickened of bronchovascular bundles, someperilymphatic nodules; thin walled lung cysts

I)iPPFE is associated with irregular apex pleural thickening with upper lobe volume loss and significant architecture distortion; anteroposterior flattening of the chest platy thorax may coexist

Where is reverse halo sign?

Response 2: We have modified the figure 8, as requested, and we have better explained the HRCT scans in the figure caption (page 19, paragraph: 5, lines 615 to 622)

Reviewer 3 Report

Comments and Suggestions for Authors

The article summarizes the diagnostic approach to idiopathic interstitial pneumonias (IIPs) from the pathologists’ point of view. The authors discuss the pathologic features of various IIPs, some of the data included in the review have been published many years ago.

1.      The article lacks the paragraph concerning combined pulmonary fibrosis and emphysema (CPFE) – especially the new recommendations published in 2022 (Cottin V. Am J Respir Crit Care Med 2022; 206: e7-e41).

2.      In differential diagnosis of NSIP and COP – hypersensitivity pneumonitis (HP) have been mentioned. The authors used the nomenclature of “chronic HP”, which have been abandoned in 2020, due to publication of recent ATS/ERS guidelines ( Raghu G et al. Am J Respir Crit Care Med 2020; 202: e36-e69). Presently HP is divided into fibrotic and non-fibrotic type.

3.      The diagnostic role of various types of tissue sampling hasn’t been sufficiently addressed by the authors. Especially the role of lung cryobiopsy in pathological diagnosis (indications, profits and inferiorities, comparison with surgical biopsy).

4.      Taking into account the journal’s profile, I would expect much more data concerning the emerging role of molecular markers in differential diagnosis of IIP (Stainer A et al. IJMS 2021; 22:6255,  Glenn LM et al. Front Med 2023;10:1174443, Cecchini MJ et al. Am J Surg Pathol 2021; 45: 871-874, Lasky JA et al. Ann Am Thorac Soc 2022; 19: 916-924)

5.      An interesting problem, that may be expanded,  is the frequent lack of agreement between radiologist and pathologist concerning the type of IIP (eg criteria of LIP recognition – Fraune C et al. Am J Surg Pathol 2023;47: 281-295) or the role of MDD in differential diagnosis of IIP (Cotin V. et al. Eur Respir Rev 2022; 31: 220003)

In summary,  I think that the review could be published, but it needs major revision:  shortening the parts of the text that contain well known data and introducing more recent knowledge concerning IIP recognition.

Author Response

Reviewer 3 The article summarizes the diagnostic approach to idiopathic interstitial pneumonias (IIPs) from the pathologists’ point of view. The authors discuss the pathologic features of various IIPs, some of the data included in the review have been published many years ago.

Comment to reviewer: Dear Reviewer, thank you very much for taking the time to review this manuscript and for your suggestions which have greatly improved the quality of the work

Comment 1:      The article lacks the paragraph concerning combined pulmonary fibrosis and emphysema (CPFE) – especially the new recommendations published in 2022 (Cottin V. Am J Respir Crit Care Med 2022; 206: e7-e41).

Response 1: We have added a paragraph named "4. Combined Pulmonary Fibrosis and Emphysema: a newly and still poorly understood clinical-pathological entity" (page 17, paragraph: 4, lines 503 to 538). This new paragraph explains the main aspects of this new entity, especially all its histopathological features.

Comment 2:      In differential diagnosis of NSIP and COP – hypersensitivity pneumonitis (HP) have been mentioned. The authors used the nomenclature of “chronic HP”, which have been abandoned in 2020, due to publication of recent ATS/ERS guidelines (Raghu G et al. Am J Respir Crit Care Med 2020; 202: e36-e69). Presently HP is divided into fibrotic and non-fibrotic type.

Response 2: We have modified the term “chronic HP” with Fibrotic Hypersensitivity Pneumonitis (fHP) (page 15, paragraph: 3.8, lines 449)

Comment 3:      The diagnostic role of various types of tissue sampling hasn’t been sufficiently addressed by the authors. Especially the role of lung cryobiopsy in pathological diagnosis (indications, profits and inferiorities, comparison with surgical biopsy).

Response 3: We have, accordingly with this suggestion, better addressed the advantages and disadvantages of different biopsy samples. We have especially discussed the advantages of lung cryobiopsy over the forceps biopsy and surgical lung biopsy and the current indications for cryobiopsy, always underlining the role of multidisciplinary discussion in choosing the type of sample (page 20, paragraph: 6, lines 639 to 664)

Comment 4:      Taking into account the journal’s profile, I would expect much more data concerning the emerging role of molecular markers in differential diagnosis of IIP (Stainer A et al. IJMS 2021; 22:6255,  Glenn LM et al. Front Med 2023;10:1174443, Cecchini MJ et al. Am J Surg Pathol 2021; 45: 871-874, Lasky JA et al. Ann Am Thorac Soc 2022; 19: 916-924)

Response 4: We have modified and expanded the paragraph named “Molecular testing as useful partner of morphology: the beginning of a new era”, highlighting the utility of the test in distinguish UIP pattern from non-UIP pattern and its impact on multidisciplinary team’s clinical decision-making in the diagnosis of IPF. We discussed about its specificity in diagnosis of UIP pattern/IPF of the molecular test and its advantages compared to histological diagnosis alone. Furthermore, we reported the impact of Envisa genomic classifier on physicians' clinical decision-making in the diagnosis and management of IPF. We have also mentioned the application of computer-based deep learning algorithms in assessment of ILD histopathology, particularly the use of algorithm to identify and quantify specific ILD features, such as fibroblastic foci, interstitial mononuclear inflammation, and intra-alveolar macrophages, in lung tissue samples (Testa, L.C.; Jule, Y.; Lundh, L.; Bertotti, K.; Merideth, M.A.; O'Brien, K.J.; Nathan, S.D.; Venuto, D.C.; El-Chemaly, S.; Malicdan, M.C.V.; et al. Automated Digital Quantification of Pulmonary Fibrosis in Human Histopathology Specimens. Frontiers in medicine, 2021, 8, 607720. doi:10.3389/fmed.2021.607720) (Mäkelä, K.; Mäyränpää, M.I.; Sihvo, H.K.; Bergman, P.; Sutinen, E.; Ollila, H.; Kaarteenaho, R.; Myllärniemi, M. Artificial intelligence identifies inflammation and confirms fibroblast foci as prognostic tissue biomarkers in idiopathic pulmonary fibrosis. Human pathology, 2021, 107, 58–68. doi:10.1016/j.humpath.2020.10.008) Finally, we reported the potential diagnostic, prognostic and predictive role of tissue biomarkers and alterations in telomere-related genes in ILDs. (Cecchini, M.J.; Tarmey, T.; Ferreira, A.; Mangaonkar, A.A.; Ferrer, A.; Patnaik, M.M.; Wylam, M.E.; Jenkins, S.M.; Spears, G.M.; Yi, E.S.; et al. Pathology, Radiology, and Genetics of Interstitial Lung Disease in Patients With Shortened Telo-meres. The American journal of surgical pathology, 2021, 45(7), 871–884. doi:10.1097/PAS.0000000000001725) (Stainer, A.; Faverio, P.; Busnelli, S.; Catalano, M.; Della Zoppa, M.; Marruchella, A.; Pesci, A.; Luppi, F. Molecular Bi-omarkers in Idiopathic Pulmonary Fibrosis: State of the Art and Future Directions. International journal of molecular sci-ences, 2021, 22(12), 6255. doi.org/10.3390/ijms22126255) (pages 20-21, paragraph: 7, lines 675 to 723)

Comment 5: An interesting problem, that may be expanded, is the frequent lack of agreement between radiologist and pathologist concerning the type of IIP (eg criteria of LIP recognition – Fraune C et al. Am J Surg Pathol 2023;47: 281-295) or the role of MDD in differential diagnosis of IIP (Cotin V. et al. Eur Respir Rev 2022; 31: 220003)

Response 5: Dear reviewer, accordingly with this suggestion, we have added the paragraph “8. Multidisciplinary team: sharing and discussion” in which we discussed about the possible lack of agreement between radiological and histological diagnoses and the fundamental role of the multidisciplinary team in the diagnostic-therapeutic management of patients with IIPs. (pages 22-23, paragraph: 8, lines 728 to 769)

In summary, I think that the review could be published, but it needs major revision:  shortening the parts of the text that contain well known data and introducing more recent knowledge concerning IIP recognition.

Round 2

Reviewer 3 Report

Comments and Suggestions for Authors

Dear Authors,

Thank you for replying to all my comments and for the introduction the new data into the text. It is a pleasure to recommend the revised form of manuscript for publication in IJMS.

My only remark is to add "HP" in line 656, due to the presence of significantly increased lymphocytosis in BALF in this ILD.